# Tolerability and Efficacy of s.c. IgG Self-Treatment in ME/CFS Patients with IgG/IgG Subclass Deficiency: A Proof-of-Concept Study

**DOI:** 10.3390/jcm10112420

**Published:** 2021-05-29

**Authors:** Carmen Scheibenbogen, Franziska Sotzny, Jelka Hartwig, Sandra Bauer, Helma Freitag, Kirsten Wittke, Wolfram Doehner, Nadja Scherbakov, Madlen Loebel, Patricia Grabowski

**Affiliations:** 1Institute of Medical Immunology, Charité–Universitätsmedizin Berlin, Corporate Member of Freie Universität Berlin and Humboldt-Universität zu Berlin, Augustenburger Platz 1, 13353 Berlin, Germany; franziska.sotzny@charite.de (F.S.); jelka.hartwig@gmail.com (J.H.); sandra.bauer@charite.de (S.B.); helma.freitag@charite.de (H.F.); kirsten.wittke@charite.de (K.W.); patricia.grabowski@charite.de (P.G.); 2Berlin Institute of Health Center for Regenerative Therapies (BCRT), Charité–Universitätsmedizin Berlin, Corporate Member of Freie Universität Berlin and Humboldt-Universität zu Berlin, Augustenburger Platz 1, 13353 Berlin, Germany; wolfram.doehner@charite.de (W.D.); nadja.scherbakov@charite.de (N.S.); 3Center for Stroke Research Berlin, Charité–Universitätsmedizin Berlin, Corporate Member of Freie Universität Berlin and Humboldt-Universität zu Berlin, Augustenburger Platz 1, 13353 Berlin, Germany; 4Research Center, Carl-Thiem-Klinikum Cottbus gGmbH, Thiemstraße 111, 03048 Cottbus, Germany; M.Loebel@ctk.de

**Keywords:** chronic fatigue syndrome, myalgic encephalomyelitis, autoimmunity, immunology, IgG replacement, IgG deficiency, biomarker

## Abstract

Background: Chronic fatigue syndrome (ME/CFS) is a complex disease frequently triggered by infections. IgG substitution may have therapeutic effect both by ameliorating susceptibility to infections and due to immunomodulatory effects. Methods: We conducted a proof of concept open trial with s.c. IgG in 17 ME/CFS patients suffering from recurrent infections and mild IgG or IgG subclass deficiency to assess tolerability and efficacy. Patients received s.c. IgG therapy of 0.8 g/kg/month for 12 months with an initial 2 months dose escalation phase of 0.2 g and 0.4 g/kg/month. Results: Primary outcome was improvement of fatigue assessed by Chalder Fatigue Scale (CFQ; decrease ≥ 6 points) and of physical functioning assessed by SF-36 (increase ≥ 25 points) at month 12. Of 12 patients receiving treatment per protocol 5 had a clinical response at month 12. Two additional patients had an improvement according to this definition at months 6 and 9. In four patients treatment was ceased due to adverse events and in one patient due to disease worsening. We identified LDH and soluble IL-2 receptor as potential biomarker for response. Conclusion: Our data indicate that self-administered s.c. IgG treatment is feasible and led to clinical improvement in a subset of ME/CFS patients.

## 1. Introduction

Chronic Fatigue Syndrome (ME/CFS) is a frequent, severe and complex disease with an estimated prevalence of around 0.5% [1]. Patients suffer from sustained exhaustion accompanied by numerous physical and mental symptoms. ME/CFS onset is typically with an infection and many patients undergo frequently recurrent infections. The underlying pathological mechanism in ME/CFS is not known so far. However, there is ample evidence of dysregulation of the immune system, and both immune activation and deficiency can be found [2,3]. There is increasing evidence, that at least in a subset of ME/CFS patients, autoimmunity contributes to disease etiology [2,4]. Autoantibodies against various antigens including neurotransmitter receptors were reported by several groups (reviewed in [2]). First clinical trials showed that immunomodulatory treatments targeting autoantibodies are effective in a subset of ME/CFS patients [5,6,7]. 

Immunoglobulin G (IgG) treatment is effective in autoantibody-mediated autoimmune diseases [8]. Four randomized controlled clinical trials of intravenous (IV) IgG therapy with monthly doses ranging from 0.5 to 2 g/kg body weight were performed more than three decades ago in ME/CFS showing inconsistent results with two positive and two negative studies [9,10,11,12,13]. IgG substitution may have therapeutic effect in ME/CFS both by ameliorating susceptibility to infections and due to immunomodulatory effects.

ME/CFS patients frequently suffer from susceptibility to both viral and bacterial infections. Studies in our own and other patient cohorts show that IgG1 and IgG3 deficiency occurs frequently in ME/CFS patients [3,11,14]. Patients with immunoglobulin deficiency had more frequently an increased rate of infections, mostly of the respiratory tract [3,15]. 

Thus far, there has been little interest of pharmaceutical companies into clinical trials in ME/CFS presumably due to the complexity of the disease and paucity of knowledge. We thus performed an investigator-initiated trial to study the feasibility and efficacy of an intermediate dose self-administered s.c. IgG treatment in ME/CFS patients. An s.c. IgG treatment regimen was chosen due to better tolerability and possibility of self-administration. The IgG dose of 0.8 g/kg body weight/month was chosen which is the maximum dose recommended by European Medicines Agency (EMA) for IgG treatment of immunodeficient patients but is expected to be effective in autoimmunity as well [8]. Further, we decided to include ME/CFS patients with mild IgG or IgG subclass deficiency and frequent infections as this group of patients may benefit from both immunomodulatory and infection-preventing effect of IgG treatment [10]. 

Our study provides evidence that self-administered s.c. IgG treatment is feasible and tolerable and can lead to clinical improvement in a subset of ME/CFS patients.

## 2. Materials and Methods

### 2.1. Study Design

This was an investigator-initiated one arm trial with support from Baxalta, a member of the Takeda group of companies. HyQvia consists of human normal IgG (10%, Kiovig^®^, Takeda, Konstanz, Germany) and recombinant human hyaluronidase (rHuPH20, Hylenex^®^, Takeda, Konstanz, Germany). The clinical study protocol is provided in the Appendix A. HyQvia was given as s.c. infusion via the Freedom pump (RMS Medical Products, Chester, NY, USA). Following pretreatment with hyaluronidase up to 300 mL of HyQvia were infused s.c. The first infusion was given at our outpatient clinic and patients were trained for self-therapy. Further infusions were given as home therapy under supervision of a home care nurse. The following doses were given:

Month 1, day 0: total 0.2 g/kg body weight per month (one infusion).

Month 2: total 0.4 g/kg body weight per month (given as one or bi-weekly infusion).

Months 3–12: total 0.8 g/kg body weight per month (given as bi-weekly infusion).

The primary objective of this study was to determine the effect of an intermediate dose of s.c. IgG on patient fatigue and physical functioning as assessed by the Chalder Fatigue Scale (CFQ) and SF-36, physical function domain, respectively. A clinical meaningful response was defined by an improvement of at least 50% of symptoms in the Chalder Fatigue Scale between the first visit and the 12-month follow-up visit. For this an improvement in at least 6 of the 11 items for minimum of one point improvement is required. It means that the composite score decreases by at least 6 points between enrollment and 12-months follow-up. For the SF-36 physical functioning, a clinically meaningful response is defined by an improvement of at least 50% of symptoms, thus the patient scores better in at least 5 of the 10 items between study enrollment and the 12-month follow-up. It means that the composite score increases by at least 25 points between enrollment and 12-months follow-up. The response analysis included only patients receiving the complete 12-months treatment. 

Secondary study objectives were to assess the tolerability of HyQvia in patients with ME/CFS, to assess the frequency and severity of infections, to identify markers for response, to assess additional symptoms by scoring the symptoms of Canadian Consensus Criteria (CCC) and COMPASS-31 and the suitability of step tracking and endothelial function as objective response parameters. Our study was designed in such a way that we wanted to obtain first evidence for efficacy as a prerequisite for a consecutive randomized placebo-controlled trial. Efficacy was defined as seeing a response in at least 5 of 15 patients included in the study. Patients receiving less than 3 months of treatment had to be replaced. Regular site monitoring visits were performed by the Clinical Research Organization GWT, Dresden. The trial was registered at www.clinicaltrialsregister.eu. EudraCT 2016-002370-12.

### 2.2. Patients

ME/CFS patients were selected who were diagnosed at the outpatient clinic for immunodeficiencies at the Institute of Medical Immunology at the Charité Universitätsmedizin Berlin (Berlin, Germany) and fulfilled the inclusion criteria. In order to obtain an unselected patient sample, all consecutive patients who are eligible for participation and willing to participate were included. The flow chart of participant disposition is shown as Figure 1. Diagnosis of ME/CFS was based on CCC [16] and exclusion of other medical or neurological diseases, which may cause fatigue. Further inclusion criteria were IgG or IgG subclass deficiency with a history of a serious bacterial or recurrent infections (≥4 infections during the last year prior to inclusion), and a disease severity according to the Bell scale of ≤50 of 100 [17]. However, in none of the patients IgG deficiency and infection history were severe enough for having an indication for IgG substitution (IgG > 5 g/l and none had IgA or IgM deficiency). It was planned to include 15 patients in the trial and replace patients receiving less than 3 months of treatment. All 17 patients who were approached agreed to participate in the trial. 

### 2.3. Assessment of Symptoms and Physical Functioning by Scores

Questionnaires were filled in by the patients at home and validated by the treating physicians together with the patients. Disease severity was determined before and after the 12 months IgG treatment by Bell scale with a score of 0 being equivalent to severest ME/CFS and a score of 100 being healthy. ME/CFS symptoms and physical functioning were further assessed by questionnaires: CFQ, SF-36 physical functioning, COMPASS-31 and CCC symptom scoring, at baseline, then monthly and up to 3 months after the IgG treatment. CFQ evaluates the extent and severity of fatigue assessing fatigue with 0 (healthy) to 33 (severe) [18]. SF-36 (Medical Outcome Study 36-Item Short Form Health Survey) measures health-related quality of life, with a score of 0 being equivalent to maximal disability and a score of 100 being healthy. COMPASS-31 questionnaire assesses autonomic symptoms with a score from 0 (healthy) to 100 (severe) [19]. The severity of symptoms was assessed based on quantification of CCC symptoms using a questionnaire developed by Fluge et al. [20]. Symptoms were classified according to a scale from 1 (no symptoms) to 10 (severe symptoms). The fatigue score was calculated as the mean of fatigue, malaise after exertion, need for rest and daily functioning, cognitive score as mean of memory disturbance, concentration ability and mental tiredness and immune score as mean of painful lymph nodes, sore throat and flu-like symptoms.

### 2.4. Laboratory Values

Standard laboratory parameters were assessed at the Charité diagnostics laboratory Labor Berlin GmbH (Berlin, Germany). Antibodies against ß2 adrenergic and M3 muscarinic acetylcholine receptors were determined by CellTrend GmbH (Luckenwalde, Germany) using ELISA technology. 

### 2.5. Functional Assessment

Peripheral endothelial function was evaluated by a pulse arterial tonometry (PAT) device (EndoPAT-2000, Itamar, Israel). Measurement was performed under standardized conditions after at least 15 min of supine rest in a quiet, air-conditioned room. Endothelial dysfunction was defined by reactive hyperemia index (RHI) ≤1.8 as described previously [21]. Steps were counted by a Vivofit^®^ activity tracker (Garmin, Germany). Mean daily number of steps was counted during one week before and after every 3 months for 15 months. Staff members performing these assessments were not involved in implementing any aspect of the intervention.

### 2.6. Statistical Analysis

Statistical data analyses were done using the software GraphPad Prism 6.0 for Windows, GraphPad Software, La Jolla California USA, www.graphpad.com. Continuous variables were expressed as median and interquartile range (IQR). Univariate comparison of two independent groups was performed using the Mann–Whitney-U test, comparison of two dependent groups was done using the Wilcoxon matched-pairs signed-rank test. A two-tailed *p*-value of <0.05 was considered statistically significant.

## 3. Results

### 3.1. Patients and Treatment

A total of 17 ME/CFS patients with a mild IgG or IgG subclass deficiency, but no indication for IgG substitution, were included in this trial (Figure 1). Patient characteristics are summerized in Table 1 (for details see Appendix A). The mean age was 46 years, 9 patients were female and 12 patients had an infection-triggered onset. We retrospectively collected the data of the types of infection from the patients records (Appendix A). Most patients reported a respiratory tract infection or primary EBV as trigger of disease. At baseline all patients had a Bell score ≤ 50. In 4 of 17 patients physicians decided to cease treatment due to adverse events as described below. In one patient treatment was discontinued at month 3 due to disease worsening (P12). A total of 12 patients received the scheduled 12-months treatment.

### 3.2. Tolerability

Patients’ adverse events are listed in Appendix A. Two patients (P1, P14) received only one or two injections and were replaced (Appendix A). Patient 1 had injection-related grade 3 headache and received only two IgG injections. Patient 14 had a grade 2 injection site reaction with an erythema of approx. 10 cm after the first IgG injection. Treatment was not continued as the patient refused to come into the outpatient clinic for the next injection due to multiple chemical sensitivity. In two patients treatment was ceased at months 3 and 6 due to adverse events (P3, P15). In patient 15 treatment was stopped at month 3 due to elevation grade 3 of the liver enzymes ALT/AST. Pretreatment ALT/AST were normal. A total of 4 weeks after cessation values had returned to grade 1. In patient P3 treatment was not continued at month 6 due to recurrent grade 2 local reaction, flu-like symptoms, headache, and abdominal pain.

In the 12 patients receiving the 12 months treatment 4 patients reported recurrent grade 2 and 2 patients grade 1 headache after injections. In three patients we observed a transient grade 1 increase in the ALT (at month 3, 6 and 9, respectively). P11 had recurrent grade 2 erythematous injection site reaction from month 9 on. All patients had mostly grade 1 flu-like symptoms and injection-site reaction.

### 3.3. Clinical Treatment Response

A total of 12 patients received IgG treatment per protocol for 12 months. The overall response in all 12 patients showed significantly decreased fatigue (measured by CFQ) and increased physical functioning (measured by SF-36) at months 6, 9 and 12, but not after the dose escalation phase at month 3 (Figure 2a,b). According to the primary response definition five of these patients (P2, 9, 11, 13, 16) had a clinical response with a decrease of at least six points in the CFQ and/or an increase of 25 points in the SF-36 physical functioning at month 12 compared to pretreatment (Figure 2c,d). Two additional patients did fulfill the primary response definition at months 6 and 9 but not at month 12 (P4, 8) (Figure 2c,d). 3 months following cessation of IgG treatment (month 15) physical functioning decreased and fatigue increased again (Figure 2a,b). As expected, we observed a close correlation between fatigue assessed by CFQ and physical functioning assessed by SF-36 (r = −0.64; *p* = 0.02). CFQ and SF36 of the three patients ceasing treatment at months 3 (P12, P15) and month 6 (P3) are shown in Appendix A.

As a secondary outcome parameter, severity of various symptoms was assessed. In the 12 patients receiving the 12 months IgG treatment an overall significant improvement in the severity of cognitive symptoms and immune symptoms could be observed at months 6, 9 and 12 (Appendix A). Improvement of the immune symptoms was already evident at month 3 and remained post IgG treatment at month 15. Muscle pain showed no overall improvement (Appendix A). The course of symptoms in the individual patients is shown in Appendix A.

All patients reported moderate to severe symptoms of autonomic dysfunction assessed by the COMPASS-31 questionnaire at baseline (median 50.9, range 12.9–73.8). Again an overall significant improvement of autonomic nervous system function from baseline to months 6 (median 37.14, range 6.99–60.06) and 9 (median 36.05, range 20.16–54.72) was observed (Appendix A).

### 3.4. Functional Assessment

We tried to objectively assess symptoms by measuring endothelial cell function and daily steps as secondary outcome parameter. The numbers of steps were assessed for a week each month by a Vivofit^®^ activity tracker (Garmin, Germany). Patient 9 could not be evaluated because the tracker of the pretreatment evaluation was lost. All responding patients at month 12 (patients 2, 11, 13 and 16) and also patients 4 and 8 with a response at month 6 and 9 walked more steps during IgG treatment (steps/day pretreatment: median 4565, range 1062–7756; at months 6: median 6067, range 3017–10411; *p* = 0.0313, Appendix A). The number of steps in the five non-responder patients (patients 5, 6, 7, 10, 17) did not increase. There was no seasonal variation in numbers of steps.

A subset of patients with ME/CFS has endothelial dysfunction [22,23]. Endothelial dysfunction defined as a diminished RHI < 1.8 was found pretreatment in 6 of 11 patients receiving 12 months of treatment. At month 15 we observed improvement of endothelial function in six of eight patients (Appendix A). No correlation of pre- or posttreatment endothelial function with clinical response was observed. 

### 3.5. Infections

In the 12 months before initiation of IgG treatment a median of six (range 4–12) mostly respiratory tract infections were reported by the patients. During the 12 months IgG treatment period 11 of 12 patients documented infections to occur less frequently and/or milder with a median of 3.5 (range 0–6) infections (Figure 3, *p* = 0.002). 

### 3.6. Assessment of Potential Biomarkers for Response and Tolerability

In the 12 patients receiving IgG treatment for 12 months we compared demographic and clinical data and laboratory values between the five responders and seven non-responders. There was no obvious difference in age, sex, Bell score and disease duration (Appendix A) nor in SF-36, CFQ, COMPASS-31 and symptom scores regarding treatment response (Appendix A). There was also no difference in these parameters when patients 4 and 8 with response at months 6 and 9 were included into the responder group. All responder had an infection-triggered onset. The five patients with a non-infectious disease onset included two patients (P1 and P3) not tolerating treatment, two non-responders and P12 in whom treatment was discontinued due to disease worsening at month 6 (Appendix A). 

We compared number of leukocytes, lymphocytes, erythrocytes, IgG and IgG subclass levels, CRP level, ANA and levels of several potential disease biomarkers (LDH, CK, soluble IL-2 receptor, IL-8, β2 AdR and M3 AchR AAB and soluble CD26) pretreatment but observed no significant differences between responders and non-responders (Appendix A). We observed, however, a trend of a higher pretreatment LDH level and lower soluble IL-2 receptor sCD25 in responders. When we included patients 4 and 8 with a response at months 6 and 9 in the responder group, responders had a significantly higher baseline serum LDH level compared to non-responders (Figure 4a). LDH levels did not decrease in the total patient cohort at month 3 during the dose escalation phase but from month 6 on (at month 9 *p* = 0.0137, Figure 4b). Further pretreatment values of the soluble IL-2 receptor were significantly lower in responder patients including patients 4 and 8 than in non-responders (Figure 4c). LDH and soluble IL-2 receptor did neither correlate with each other nor with disease severity or symptoms. IgG levels during treatment increased from median 8.5 g/L to maximum median 15.4 g/L at month 9 (Figure 4d). IgG and IgG subclass levels pretreatment did not correlate with response to treatment (Appendix A). In the study by Lloyd et al. higher lymphocyte count pretreatment was associated with response to IgG treatment. Here, we could not confirm this observation (Appendix A) [10].

When comparing patients not tolerating treatment to those receiving 12 months of treatment we observed no differences in clinical parameters. Interestingly, we observed a lower level of the biomarker sCD26 in the five patients who did not tolerate treatment or had early disease worsening compared to patients who completed IgG treatment (Figure 4e).

## 4. Discussion

In this study we provide evidence that self-administered s.c. IgG treatment is feasible and improved symptoms and physical functioning in a subset of ME/CFS patients with a mild IgG or IgG subclass deficiency. Our results are in line with previous randomized controlled clinical trials (RCT) studies.

Four RCTs of IV immunoglobulin replacement therapy with monthly doses ranging from 0.5–2 g/kg body weight were performed more than three decades ago with two positive and two negative studies [9,10,11,12,13]. In the positive Australian study by Lloyd et al. 10 of 23 (43%) patients receiving IV IgG 2 g/kg on a monthly basis for 3 months but only three of the 26 (12%) placebo recipients responded with a substantial reduction in their symptoms and recommencement of work and activities [10]. In the negative follow-up four-arm study with 99 patients receiving one of three doses of immunoglobulin (0.5, 1, or 2 g/kg) or a placebo solution (1% albumin), all patients showed a similar improvement in their functional capacity [12]. In the other positive study by Rowe et al. in which 71 adolescents received either three infusions of 1 g/kg given one month apart or 1% albumin, both groups had an improved functional score, which was significantly higher in the IgG treated patients at 6 months [13]. In the negative US study Peterson et al. had treated 28 patients in a RCT with IV IgG 1 g/kg or placebo [11]. Both patient groups reported improved health perception. Taken together, improvement of IgG treated patients was reported in all four trials, but also in the placebo group in two of these four trials. This is in line with the results of the recently published multicenter RCT Rituximab study performed in Norway in which around a third of patients reported improvement in both the rituximab and the placebo infusion group [24]. Response definition was, however, less strict, with fatigue improvement for 8 consecutive weeks over a period of 24 months.

How should these findings be translated into a future clinical IgG study in ME/CFS? Our study was designed to observe efficacy in at least 5 of 15 patients as a prerequisite for a consecutive randomized placebo-controlled trial. According to the primary response definition we reached this aim with 5 of 15 patients receiving at least 3 months of treatment having a clinical response. A dose-escalation phase can help in a consecutive randomized placebo-controlled trial to control for placebo effects. In the 12 patients receiving 12 months of treatment we observed overall significantly improved CFQ and SF-36 scores at months 6, 9 and 12, but not after the dose escalation phase at month 3. Objective parameters of response would be desirable in a consecutive randomized placebo-controlled trial. We evaluated the suitability of step counting in our response assessment and observed an increase in number of steps in responder but not non-responder patients. 

Further, it is important to implement biomarker for response. IgG deficiency and higher lymphocyte counts were associated with response to treatment in the study by Lloyd et al. [10]. In our study, IgG or IgG subclass deficiency was an inclusion criterion and lymphocyte counts were similar between responders and non-responders. We studied several immune and metabolic biomarkers which were shown to be altered in a subgroup of ME/CFS patients in previous studies [25,26,27,28]. Evidence from our study suggests that pretreatment elevated LDH and lower soluble IL-2 receptor levels are a potential response marker. Further we observed a significant decrease in LDH during IgG treatment. How could this be explained? The LDH is an enzyme catalyzing the conversion of pyruvate to lactate and may indicate a preferential energy production via glycolysis rather than the more efficient oxidative phosphorylation as described in ME/CFS [29]. This may result in impaired function of immune cells and lower soluble IL-2 receptor levels, too. Thus, the elevated LDH and lower soluble IL-2 receptor levels may be biomarkers for an impaired metabolism in ME/CFS.

What are the potential mechanisms of IgG treatment in ME/CFS? The infection control by IgG is one possibility as many ME/CFS patients suffer from frequent and long-lasting infections which frequently result in disease aggravation [3]. IgG deficiency is associated with more infections and is frequent in ME/CFS and was reported also in approximately half of the patients in the studies by Lloyd and Peterson [11,12]. IgG deficiency was associated with response to treatment in the study by Lloyd. Therefore, we included in our study only patients with mild IgG or IgG subclass deficiency and recurrent infections. Patients reported less frequent and milder infections during IgG treatment, which may have an effect on amelioration of disease severity. IgG in a dose of 0.8 g/kg body weight s.c. can, however, have immunomodulatory effects as well and it is well known that IgG treatment is effective in autoantibody-mediated autoimmune diseases. There is evidence that infection-triggered ME/CFS is an autoimmune disease [2]. In line with this notion all responder in our study had an infection-triggered onset. In three of the previous RCT studies information on the type of onset is provided with the majority of patients reporting an infection-triggered onset (76–97%) [10,11,12]; however, the authors did not provide information if this had an impact on response. There is evidence of clinical efficacy of other immunomodulatory treatments targeting autoantibodies in ME/CFS including previous rituximab trials with a dose of 500 mg/m^2^, immunoadsorption, and endoxan [5,6,7]. A total of 3 months following cessation of IgG treatment physical functioning assessed by SF-36 decreased again. Remission times of IgG treatment in autoimmune diseases can be short [8]. Thus, it would be desirable to treat ME/CFS patients for more than 12 months with IgG in a consecutive study.

Treatment with s.c. IgG is usually well tolerated in primary immunodeficiency (PID) [30]. Compared to PID patients we observed side-effects to occur frequently in patients with ME/CFS including headaches and liver enzyme elevations. This is in line with the study by Lloyd et al. who reported that constitutional symptoms occurred with 53 of the 65 IgG (82%) but only 19 of the 78 placebo infusions (24%) [10]. Transient elevation of ALT levels developed in eight IgG recipients but only one placebo recipient [10]. Similarly, in the study by Peterson et al. headaches were reported more frequently in the IgG group [11]. Thus, a higher frequency of adverse events should be taken into account when treating patients with ME/CFS with IgG. 

We observed diminished sCD26 to be associated with intolerability of IgG treatment and early disease deterioration. sCD26 was reported in two previous studies to be diminished in a subset of ME/CF patients [25,31]. sCD26 or sDPP4 is well known as an enzyme cleaving glucagon, but also various immune mediators including chemokines and bradykinin [32,33]. It may well be that diminished levels of sCD26 are associated with elevated levels of such immune mediators which may result in more side effects of IgG therapy. Several patients reported better tolerability of the infusions with improvement of their ME/CFS symptoms during treatment. Similarly, more than the expected number of side-effects were observed in the rituximab trial [24].

## 5. Conclusions

Taken together, our study has several limitations, including a small patient number and a lack of a control arm. The strength of this study is to show the feasibility of a dose escalation s.c. IgG home treatment in ME/CFS patients. Furthermore, it provides first evidence for efficacy of an intermediate dose s.c. IgG treatment and potential biomarkers for response. This warrants an RCT study.

## Figures and Tables

**Figure 1 jcm-10-02420-f001:**
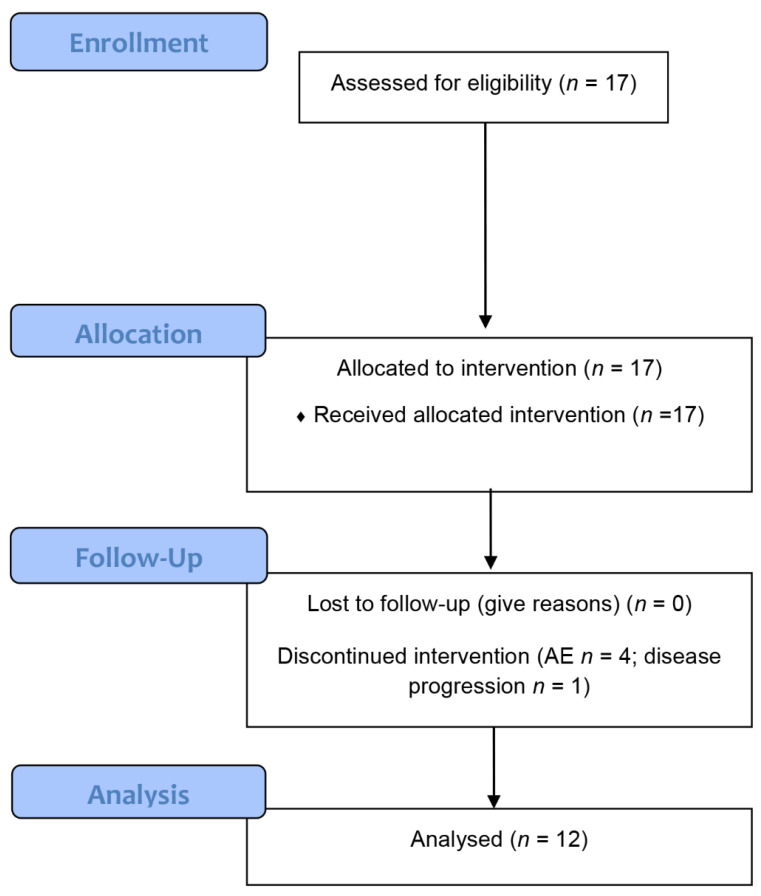
Flow chart of participant disposition. AE = adverse events.

**Figure 2 jcm-10-02420-f002:**
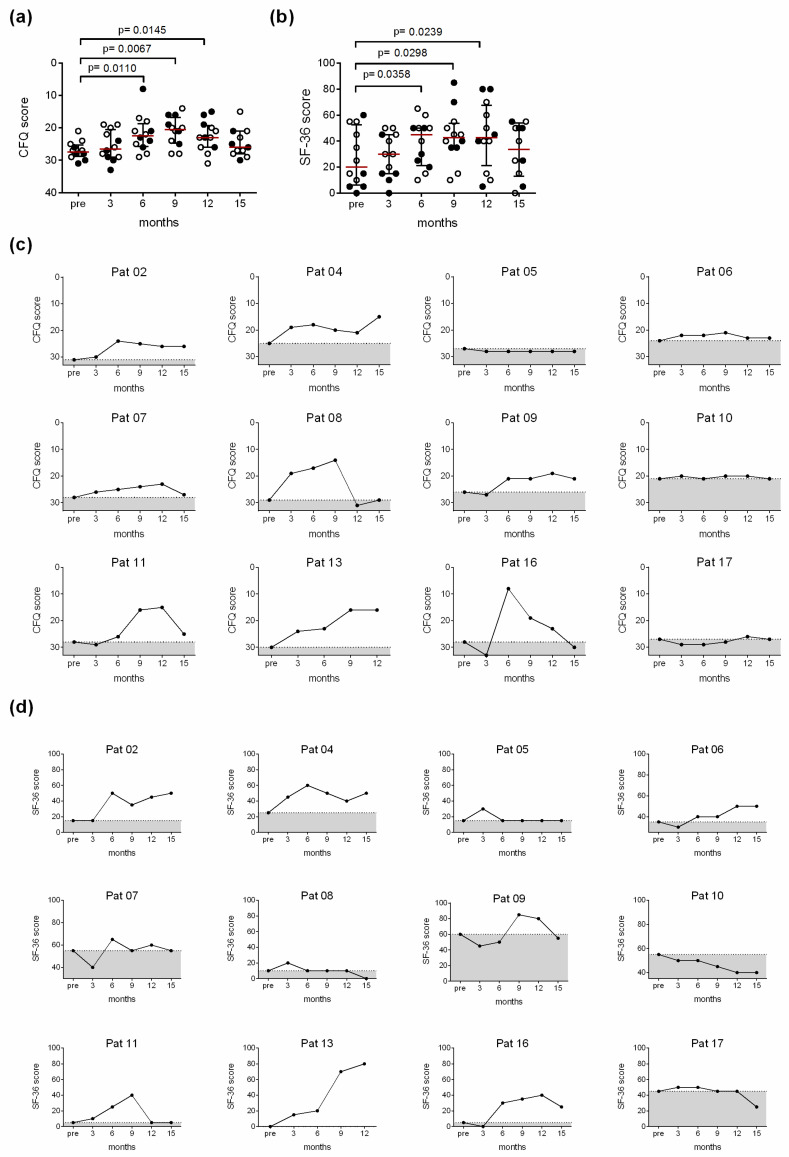
(**a**) The Chalder Fatigue Scale (CFQ) and the (**b**) SF-36 physical functioning of all patients receiving 12 months of treatment before (pre), during (months 3–12) and 3 months after the treatment (month 15) is shown. An overall significant improvement of the fatigue (CFQ) and physical functioning (SF-36) during the IgG treatment was observed (responder indicated as filled circles). A two-tailed Wilcoxon matched-pairs signed-rank test was performed for statistical analysis. The course of individual patients is shown in (**c**) for CFQ and (**d**) for SF-36. (CFQ Score, healthy: 0; SF-36 Score, healthy: 100).

**Figure 3 jcm-10-02420-f003:**
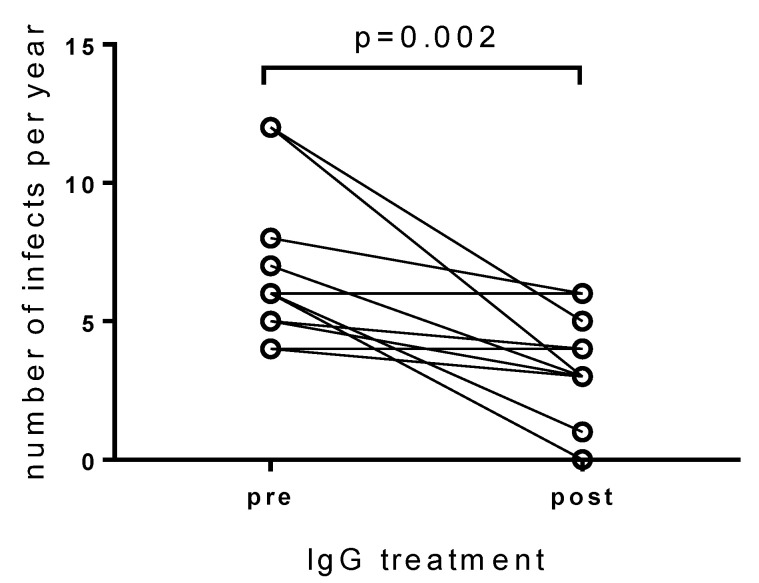
Numbers of infections per year in the 12 months before (pre) and 12 months after (post) initiation of IgG treatment are shown. Numbers of infections significantly decreased during the IgG treatment. A two-tailed Wilcoxon matched-pairs signed-rank test was performed for statistical analysis.

**Figure 4 jcm-10-02420-f004:**
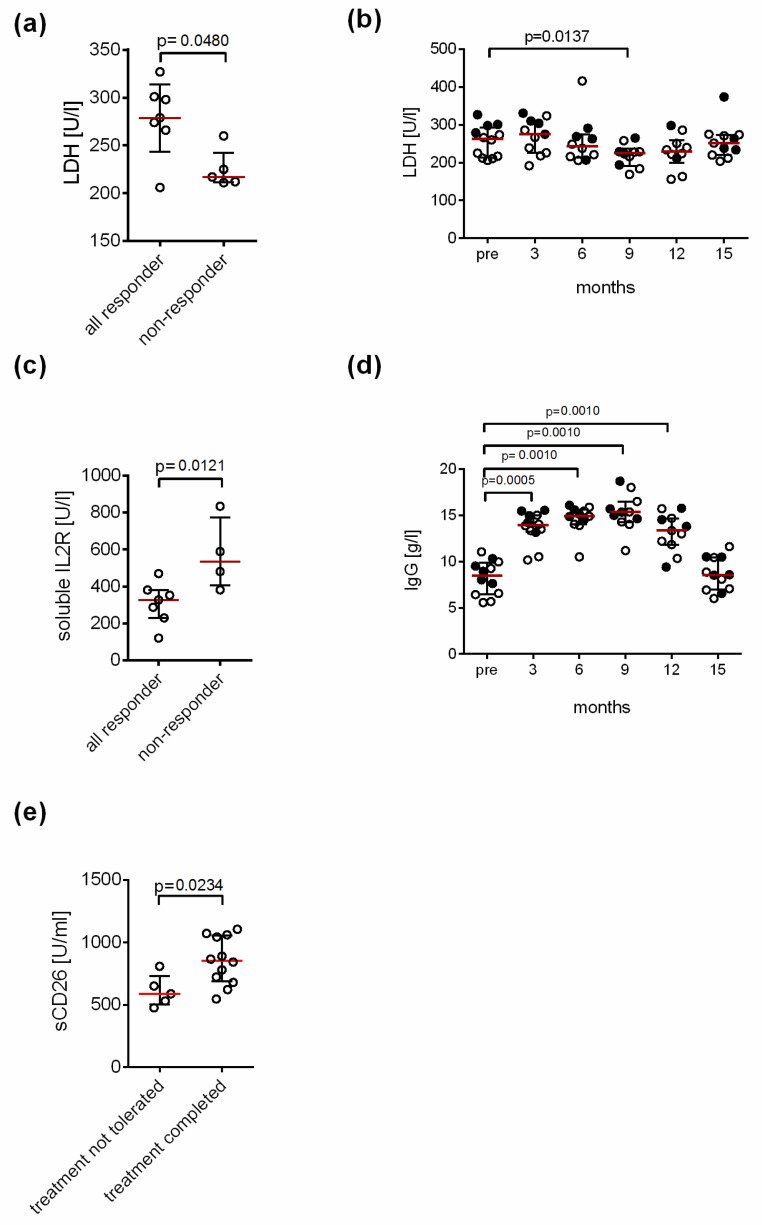
Association of biomarker with response and tolerability. Median and interquartile range pretreatment level of (**a**) LDH and (**c**) the soluble IL-2 receptor of responding patients including patients with a response at month 6 and 9 (*n* = 7) compared to non-responders (*n* = 5) are shown. Responder had significant higher LDH and lower soluble IL-2 receptor pretreatment level. Two tailed Mann–Whitney-U test was performed for statistical analysis. Median and interquartile range (**b**) LDH level and (**d**) IgG level of the patients before (pre), during (months 3–12) and 3 months after the treatment (month 15) are shown (responder indicated as filled circles). A two-tailed Wilcoxon matched-pairs signed-rank test was performed for statistical analysis. (**e**) Median and interquartile range of sCD26 level of patients not tolerating treatment (*n* = 5) compared to those receiving 12 months of treatment (*n* = 12) are plotted. Participants who did not tolerate the treatment had significant lower sCD26 levels. Two tailed Mann–Whitney-U test was performed for statistical analysis.

**Table 1 jcm-10-02420-t001:** Patient characteristics.

	Study Cohort (*n* = 17)
sex (f/m)	9/8
age in years (median (range))	46 (18–70)
age at disease onset in years (median (range))	36 (15–61)
Bell score (median (range))	30 (20–50)
infection-triggered onset	*n* = 12

## Data Availability

The data presented in this study are available on reasonable request from the corresponding author.

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
