# Peer review of "Tolerability and Efficacy of s.c. IgG Self-Treatment in ME/CFS Patients with IgG/IgG Subclass Deficiency: A Proof-of-Concept Study"

_jcm, 2021, doi:10.3390/jcm10112420_

Round 1

Reviewer 1 Report

This is an original research paper dealing with the feasibility and efficacy of self-administered subcutaneous gamma globulin (IgG) treatment in patients with ME/CFS. The authors aimed to obtain a preliminary assessment of patient response and outcome to determine whether findings would warrant a comprehensive, large scale randomized placebo-controlled clinical trial.

The subject of this study is important for researchers and medical professionals in the ME/CFS field, as it addresses a treatment modality that generated inconclusive results in the past. These findings point to mechanisms that could possibly resolve previous inconsistencies.

There are, however, a few weak points, some of which are discussed by the authors, and others that could be improved upon in this reviewer’s opinion: First of all the low number of participants (n=17 of which 12 only could be evaluated after 12 months) and secondly, the absence of a control group.

This manuscript requires minor revisions, specifically:

Page 8, line 203: The headline should read RESULTS, not Materials and Methods

Page 10, Section 3.3 Clinical Treatment response: Start the paragraph with Figure 2a,b description, then Figure 2c,d. Add a comment on the 15 month decline.

Page 13. Section 3.4 Functional assessment: The authors are comparing the number of daily steps at two timepoints 6 months apart. Is there a seasonal variation in number of steps? so that participants walk less during winter compared to summer?

Page 13, Section 3.5 Infections: A figure or a table illustrating the observed decrease in infections with statistical analysis (infections 12 months prior vs infections 12 months after treatment) would better convey the finding.

Page 14, Lines 313-316: Noteworthy, all responders had an infectious disease onset, while non-responders did not. Please expand on what type of infectious or non-infectious trigger was present. Could the presence of an ‘infectious-triggered’ subgroup account for the discrepancy between older study findings regarding IgG treatment? If this is tractable from the published literature, please include in the discussion.

Page 15, Lines 335-337, Table S8: Please include an extra column with reference values for each parameter.

Page 15, Lines 335-337: The authors claim that there was no significant clinical parameter differentiating patients not tolerating treatment, but as explained in the above comment, the prior infectious trigger may actually be that significant parameter.

Minor comments:

Page 3, Lines 71-72: “despite limited resources”. It is this reviewer’s opinion that although understandable as a limiting factor, resource availability is not relevant in the evaluation of the scientific merit of this research. Maybe authors should avoid including such information.

Page 4, Line 107: “determine the efficacy” The efficacy of an intervention cannot be determined without the appropriate control groups. Authors should consider rephrasing. Suggestion: “Determine the patient response after…”

Page 17, Lines 362-365: If possible, the authors should insert a piece of discussion on how their findings about infectious onset may account for the conflicting previous findings, if that can be deducted from published information in these previous studies.

Grammatical/syntactical mistakes:

line 53-54: “efficacy of.. to be effective” The first efficacy of should be deleted, the immunomodulatory treatments are effective, not their efficacy

Line 107: “The primary endpoint” The authors are describing the primary objective of the study, not its endpoints.

Line 246-247: “improved fatigue” Fatigue is not improved, it is lessened or decreased maybe?

Line 302: “occur less frequent” Occur less frequently or be less frequent

Author Response

Dear Reviewer 1,

We thank you for your comments and suggestions and respond as follows:

Reviewer: Page 8, line 203: The headline should read RESULTS, not Materials and Methods

Response: This is corrected in the revised version of our manuscript.

Reviewer: Page 10, Section 3.3 Clinical Treatment response: Start the paragraph with Figure 2a,b description, then Figure 2c,d. Add a comment on the 15 month decline.

Response: We adopted our manuscript according this suggestion (page 10, section 3.3 Clinical Treatment response, line: 253)

Reviewer: Page 13. Section 3.4 Functional assessment: The authors are comparing the number of daily steps at two timepoints 6 months apart. Is there a seasonal variation in number of steps? so that participants walk less during winter compared to summer?

Response: We added the following sentence (page 14, line 320): There was no seasonal variation in numbers of steps.

Reviewer: Page 13, Section 3.5 Infections: A figure or a table illustrating the observed decrease in infections with statistical analysis (infections 12 months prior vs infections 12 months after treatment) would better convey the finding.

Response: We added a new Figure 3 to our manuscript illustrating the observed decrease in infection (page 14, line 334)

Reviewer: Page 14, Lines 313-316: Noteworthy, all responders had an infectious disease onset, while non-responders did not. Please expand on what type of infectious or non-infectious trigger was present.

Response: We added the information on the type of infectious disease onset in the new supplementary table S2.

Reviewer: Could the presence of an ‘infectious-triggered’ subgroup account for the discrepancy between older study findings regarding IgG treatment? If this is tractable from the published literature, please include in the discussion.

Response: The type of onset was reported in 3 of the 4 previous IgG studies with the majority of participants reporting an infection-triggered disease onset, thus this does not explain differences in outcome. We added this information in the discussion (page 20, line 475).

Reviewer: Page 15, Lines 335-337, Table S8: Please include an extra column with reference values for each parameter.

Response: We included the reference values in the new table S9.

Reviewer: Page 15, Lines 335-337: The authors claim that there was no significant clinical parameter differentiating patients not tolerating treatment, but as explained in the above comment, the prior infectious trigger may actually be that significant parameter.

Response: When comparing tolerability between the patient group with and without infectious onset there was a trend to significance. Due to the small patient number, we think we should however, be cautious to draw such a conclusion.                                                  

Reviewer: Page 3, Lines 71-72: “despite limited resources”. It is this reviewer’s opinion that although understandable as a limiting factor, resource availability is not relevant in the evaluation of the scientific merit of this research. Maybe authors should avoid including such information.

Response: We removed this comment.

Reviewer: Page 4, Line 107: “determine the efficacy” The efficacy of an intervention cannot be determined without the appropriate control groups. Authors should consider rephrasing. Suggestion: “Determine the patient response after…”

Response: We adopted this according to the reviewer’s suggestion.

Reviewer: Page 17, Lines 362-365: If possible, the authors should insert a piece of discussion on how their findings about infectious onset may account for the conflicting previous findings, if that can be deducted from published information in these previous studies.

Response: We added the following information in the discussion (page 20, line 475).

“In line with this notion all responder in our study had an infection-triggered onset. In three of the previous RCT studies information on the type of onset is provided with the majority of patients reporting an infection-triggered onset (76-97%) [[10-12]], however, the authors did not provide information if this had an impact on response.”

We thank the reviewer for pointing out grammatical/syntactical mistakes. We corrected them.

Best regards,

Carmen Scheibenbogen und Franziska Sotzny

on behalf of all authors

Reviewer 2 Report

Paper: Scheibenbogen et al. Tolerability and efficacy of a s.c. IgG self-treatment in patients with ME/CFS: a proof-of-concept study 

The authors present av small series of cases with ME/CFS and IgG or IgG subclass deficiency. As the authors state in the conclusion the study has several limitations including a small patient number and a lack of control arm. There are more limitations. 

#1 

ME/CFS is a diagnosis that probably has several different pathophysiologic mechanisms. As in most systemic diseases the immune system plays an important role. In this study patients with immunedeficiencies have been studied. This should come to light also in the title. The immune deficiency should be better defined – it is this subgroup of patients that there could be an indication for IgG-substitution. To treat ME/CFS in general with immune globulins is not advocated today. 

#2 

Patients with immune deficiencies and frequent infections probably have many   of the symtoms of ME/CFS even without that diagnosis. Those symptoms will probably also decrease after treatment with IgG. This should be discussed. Is it ME/CFS or the immunedeficiency that is treated? 

#3 

CCC or ICC? 

Canadian consensus criteria usually refers to Carruthers et al 2003. In 2011 these criteria were updated (Carruthers et al 2011) and these criteria are usually referred to as International Concensus Criteria, ICC 

#4 

The authors found changes in LDH and soluble IL-2 receptor levels. It seems that a lot of biomarkers were analysed. Hundreds? A lot of other measurements were performed. It should be stated how many. If analysing hundreds of variables in such a few patients there is a considerable risk of mass significance. It could be interesting to mention these finding in a pilot study in order to in future studies test these hypotheses. These findings in this study does not motivate all the disussions on this. 

#5 

CFQ versus SF-36, figure 3 

That there is a correlation between CFQ and SF-36 would be expected and there is no need to present this in a figure  

#6 

Adverse events. 

Adverse events were reported all patients mostly grade 1 flue-like symptoms and injection-site reaction. Headache was common but also effects on liver enzymes. Five patients discontinued the treatment. This high frequency of adverse events should be taken into account when treating ME/CFS-patients with immunoglobuliner. 

#7 

The heading av row 203 should be ”Results”? 

#8 

Table1 

These patient characteristics could be presented more shortly in text in stead of the large table. 

#9 

In general the results are worth to publish, but the authors analyse data as if was a larger study. The paper should be condensed and should be seen as a pilot study. 

Author Response

Dear Reviewer 2,

We thank you for your comments and suggestions and respond as follows:

 #1 

ME/CFS is a diagnosis that probably has several different pathophysiologic mechanisms. As in most systemic diseases the immune system plays an important role. In this study patients with immunedeficiencies have been studied. This should come to light also in the title.

Response: We adopted the title of our manuscript according to this reviewer`s suggestion.

The immune deficiency should be better defined – it is this subgroup of patients that there could be an indication for IgG-substitution. To treat ME/CFS in general with immune globulins is not advocated today. 

Response: In none of the patients IgG deficiency and infection history were severe enough for having an indication for IgG substitution and none had IgA or IgM deficiency.

We added this information in the Material and Methods, 2.2 (Page 6, line 155).

 #2 

Patients with immune deficiencies and frequent infections probably have many of the symptoms of ME/CFS even without that diagnosis. Those symptoms will probably also decrease after treatment with IgG. This should be discussed. Is it ME/CFS or the immunodeficiency that is treated? 

Response: We discussed this in the revised version of the manuscript (page 20, line 471)

#3 

CCC or ICC? 

Canadian consensus criteria usually refers to Carruthers et al 2003. In 2011 these criteria were updated (Carruthers et al 2011) and these criteria are usually referred to as International Concensus Criteria, ICC

Response: We thank the reviewer for his comment. For diagnosis, we used the Canadian consensus criteria and corrected the reference to Carruthers et al 2003. Furthermore, we updated the first reference to the more recent paper: Valdez et al., 2018

 #4 

The authors found changes in LDH and soluble IL-2 receptor levels. It seems that a lot of biomarkers were analysed. Hundreds? A lot of other measurements were performed. It should be stated how many. If analysing hundreds of variables in such a few patients there is a considerable risk of mass significance. It could be interesting to mention these finding in a pilot study in order to in future studies test these hypotheses. These findings in this study does not motivate all the disussions on this. 

Response: A total of 7 potential biomarkers were analysed. The analysed parameters are now listed in results 3.6, Assessment of potential biomarkers for response and tolerability (page 15, line: 355) and are all listed in table S9.

#5 

CFQ versus SF-36, figure 3 

That there is a correlation between CFQ and SF-36 would be expected and there is no need to present this in a figure  

Response: We removed figure 3.

 #6 

Adverse events. 

Adverse events were reported all patients mostly grade 1 flue-like symptoms and injection-site reaction. Headache was common but also effects on liver enzymes. Five patients discontinued the treatment. This high frequency of adverse events should be taken into account when treating ME/CFS-patients with immunoglobuliner. 

Response: We have added this point in the discussion, (page 21, line 497)

 #7 

The heading av row 203 should be ”Results”? 

Response. Thank you, we corrected the heading.

#8 

Table1 

These patient characteristics could be presented more shortly in text instead of the large table. 

Response: We included a shortened version of the table 1 in the manuscript. As reviewer 1 was interested on more patient specific information, we decided to put a revised version of the table 1 with the detailed patient characteristics in the supplements (new table S2)

#9 

In general the results are worth to publish, but the authors analyse data as if was a larger study. The paper should be condensed and should be seen as a pilot study. 

Response: We understand this opinion. However, the other reviewer asked for even more details. We would, however, be willing to shorten the manuscript if required by JCM.

Best regards,

Carmen Scheibenbogen und Franziska Sotzny

on behalf of all authors

This manuscript is a resubmission of an earlier submission. The following is a list of the peer review reports and author responses from that submission.